# Alkaline-SDS cell lysis of microbes with acetone protein precipitation for proteomic sample preparation in 96-well plate format

Yan Chen[1,2,3], Jennifer W. Gin[1,2,3], Ying Wang[4], Markus de Raad [4], Stephen Tan[1,2,3], Nathan J. Hillson[1,2,3], Trent R. Northen[2,4], Paul D. Adams[2,5,6], Christopher J. Petzold [1,2,3]*

1 Biological Systems and Engineering Division, Lawrence Berkeley National Laboratory, Berkeley, California, United States of America, 2 DOE Joint BioEnergy Institute, Emeryville, California, United States of America, 3 DOE Agile BioFoundry, Emeryville, California, United States of America, 4 Environmental Genomics and Systems Biology Division, Lawrence Berkeley National Laboratory, Berkeley, California, United States of America, 5 Department of Bioengineering, University of California Berkeley, Berkeley, California, United States of America, 6 Molecular Biophysics and Bioimaging Division, Lawrence Berkeley National Laboratory, Berkeley, California, United States of America

* cjpetzold@lbl.gov

**Data Availability Statement:** The LCMS analysis raw data have been deposited to the ProteomeXchange Consortium data depository at

## Abstract

Plate-based proteomic sample preparation offers a solution to the large sample throughput demands in the biotechnology field where hundreds or thousands of engineered microbes are constructed for testing is routine. Meanwhile, sample preparation methods that work efficiently on broader microbial groups are desirable for new applications of proteomics in other fields, such as microbial communities. Here, we detail a step-by-step protocol that consists of cell lysis in an alkaline chemical buffer (NaOH/SDS) followed by protein precipitation with high-ionic strength acetone in 96-well format. The protocol works for a broad range of microbes (*e.g.*, Gram-negative bacteria, Gram-positive bacteria, non-filamentous fungi) and the resulting proteins are ready for tryptic digestion for bottom-up quantitative proteomic analysis without the need for desalting column cleanup. The yield of protein using this protocol increases linearly with respect to the amount of starting biomass from 0.5–2.0 OD*mL of cells. By using a bench-top automated liquid dispenser, a cost-effective and environmentally-friendly option to eliminating pipette tips and reducing reagent waste, the protocol takes approximately 30 minutes to extract protein from 96 samples. Tests on mock mixtures showed expected results that the biomass composition structure is in close agreement with the experimental design. Lastly, we applied the protocol for the composition analysis of a synthetic community of environmental isolates grown on two different media. This protocol has been developed to facilitate rapid, low-variance sample preparation of hundreds of samples and allow flexibility for future protocol development.

http://www.proteomexchange.org/. They are publicly accessible with the dataset identifier PXD039268.

**Funding:** The funders had and will not have a role in study design, data collection and analysis, decision to publish, or preparation of the manuscript. The United States Government retains and the publisher, by accepting the article for publication, acknowledges that the United States Government retains a non-exclusive, paid-up, irrevocable, worldwide license to publish or reproduce the published form of this manuscript, or allow others to do so, for United States Government purposes. The views and opinions of the authors expressed herein do not necessarily state or reflect those of the United States Government or any agency thereof. Neither the United States Government nor any agency thereof, nor any of their employees, makes any warranty, expressed or implied, or assumes any legal liability or responsibility for the accuracy, completeness, or usefulness of any information, apparatus, product, or process disclosed, or represents that its use would not infringe privately owned rights. The proof-of-concept work with mono-cultures and resources were part of the Joint BioEnergy Institute (JBEI; http://www.jbei.org), the extension of the procedure and identification of the sources of error were part of the Agile BioFoundry (ABF; http://agilebiofoundry.org), and the application to synthetic microbial communities was part of Ecosystems and Networks Integrated with Genes and Molecular Assemblies (ENIGMA; http://enigma.lbl.gov), a Science Focus Area led by Lawrence Berkeley National Laboratory. This work was supported by the U.S. Department of Energy, Office of Science, Office of Biological & Environmental Research (JBEI and ENIGMA) and through the Bioenergy Technology Office within the DOE Office of Energy Efficiency and Renewable Energy (ABF) through contract DE-AC02-05CH11231 between Lawrence Berkeley National Laboratory and the U. S. Department of Energy.

**Competing interests:** N.J.H. has financial interests in TeselaGen Biotechnologies, and Ansa Biotechnologies. The other authors have declared that no competing interests exist. This does not alter our adherence to PLOS ONE policies on sharing data and materials.

## Introduction

As new applications of protein analysis increase there is a concurrent impact on analytics, such as bottom-up, quantitative proteomics, to assay the samples. The impact of this is felt most intensely in biotechnology where the construction of hundreds or thousands of samples for testing is routine [1] and for analyses where the structure of microbial communities of diverse organisms must be assayed [2]. Consequently, it is crucial to improve the quality and throughput of sample preparation methods that facilitate such assays. Recently, multiple groups have automated various steps (*e.g.*, cell lysis, protein precipitation/isolation, protein quantification, tryptic digestion) of the common bottom-up proteomic sample preparation methods for various types of samples. There are a variety of automated protocols for the tryptic digestion and peptide cleanup steps [3–7] because once the protein is purified from the cells or matrices the digestion and cleanup steps are quite similar regardless of the original source.

In contrast, there are fewer examples of automation protocols for cell lysis or protein precipitation/isolation because the source of the protein sample greatly impacts those methods and commonly-used mechanical lysis techniques (*e.g.*, sonication, bead beating, French press) are challenging to automate. But, chemical lysis methods by using detergents or surfactants (*e.g.*, SDS, Triton) [8–11], organic solvents [12, 13], acidification [14], chelating agents (EDTA), and chaotropic agents (*e.g.*, urea, guanidine) are more amenable to automation. Recently, automated methods for FASP [8, 15, 16], SP3 [9], and several commercial options (*e.g.*, iST, S-Trap tips) leveraging these methods have been developed in plate format. These types of protocols effectively reduce both the total and hands-on time for sample preparation [11] but can become quite costly for large numbers of samples. Alternatively, cell lysis by using detergents under alkaline conditions coupled with purification by protein precipitation are rapid, easily automated, and cost effective. Work by Doucette and co-workers detail how high-yield (~98%) protein precipitation can be achieved in only two minutes by using a high ionic strength 80% acetone solution [17–19]. Here, we describe a rapid protocol for samples in 96-well plate format including cell lysis, benzonase digestion of the nucleic acids, and precipitation of the proteins with the salt-acetone method [17] that is applicable to monocultures and microbial communities. Even though SDS is used for lysis, protein precipitation and subsequent washing steps efficiently remove the SDS from the samples, consequently, clean-up steps are not necessary prior to LC-MS analysis. After resuspension of the proteins, we quantify the amount of protein extracted and show a linear increase of protein released from *Escherichia coli*, *Pseudomonas putida*, *Streptomyces albus*, *Corynebacterium glutamicum*, *Saccharomyces cerevisiae*, and *Rhodosporidium toruloides*. Next, we used a previously described automation protocol [20] to normalize the amount of protein and set up tryptic digestion and then analyzed the peptides with a short 10-minute gradient LC-MS/MS data independent acquisition (DIA) method [21] for both single organisms and mixtures of organisms. Lastly, we analyzed a synthetic community of five bacteria (four Gram-negative, one Gram-positive) isolated from the environment [22] to test the methods for composition analysis by quantifying the contribution of each of the species to the total biomass. This protocol, combined with previously established automated protein quantification and protein normalization protocols, provides a rapid, cost-effective method to prepare LC-MS proteomic samples from bacteria and non-filamentous fungi cell cultures.

## Materials and methods

The protocol described in this peer-reviewed article is published on protocols.io, https://dx.doi.org/10.17504/protocols.io.6qpvr6xjpvmk/v1 and is included for printing as (S1 File) with this article.

## Expected results

The alkaline-SDS cell lysis with acetone protein precipitation protocol (S1 File) is composed of cell lysis with a mixture of 200 mM NaOH and 1% sodium dodecyl sulfate (SDS), neutralization with HCl, benzonase treatment to remove nucleic acids, followed by the addition of acetone (80% by volume) to precipitate the proteins. The protocol takes approximately 30 minutes to process one full 96-well plate of samples, including centrifugation steps. The protocol time is scalable to the number of samples to be processed. Application of this protocol with an automated liquid dispenser offers cost-savings by reducing the number of pipette tips used in the protocol and minimizing reagent waste due to extremely low dead volumes. The total extracted protein amount using the described protein extraction protocol varies on our tested microorganisms, but the protein amounts extracted from them all show a linear relationship to the amount of biomass used in the test (Fig 1). With two OD*mL (~2 ×10$^9$ cells) of biomass, we obtained up to 88 μg and 131 ug total proteins from Gram-positive bacteria species of *C. glutamicum* and *S. albus* (Fig 1A), 417 ug and 258 ug total proteins from Gram-negative bacteria species of *E. coli* and *P. putida* (Fig 1B), and 400 ug and 174 ug total proteins from non-filamentous fungi species of *S. cerevisiae* and *R. toruloides* (Fig 1C). The amount of protein extracted from the microbes will depend on the species of interest, consequently, the starting amount of biomass may need to be adjusted for a given application. The protein amounts are sufficient for typical nano- and standard-flow LC-MS data acquisition methods and can easily be adjusted for applications requiring larger amounts of protein such as deep-proteome analysis by fractionation or post-translational modification characterization experiments. The upper limit on the amount of biomass that can be processed with this protocol is limited by the amount of SDS-alkaline based lysis buffer that can be added to the PCR plate (~25 μL). For applications that require larger amounts of protein, such as multi-dimensional chromatography, the protocol can easily be adapted to extractions in deep-well plates with correspondingly more lysis buffer and acetone. The protocol can also be scaled down to approximately 0.25

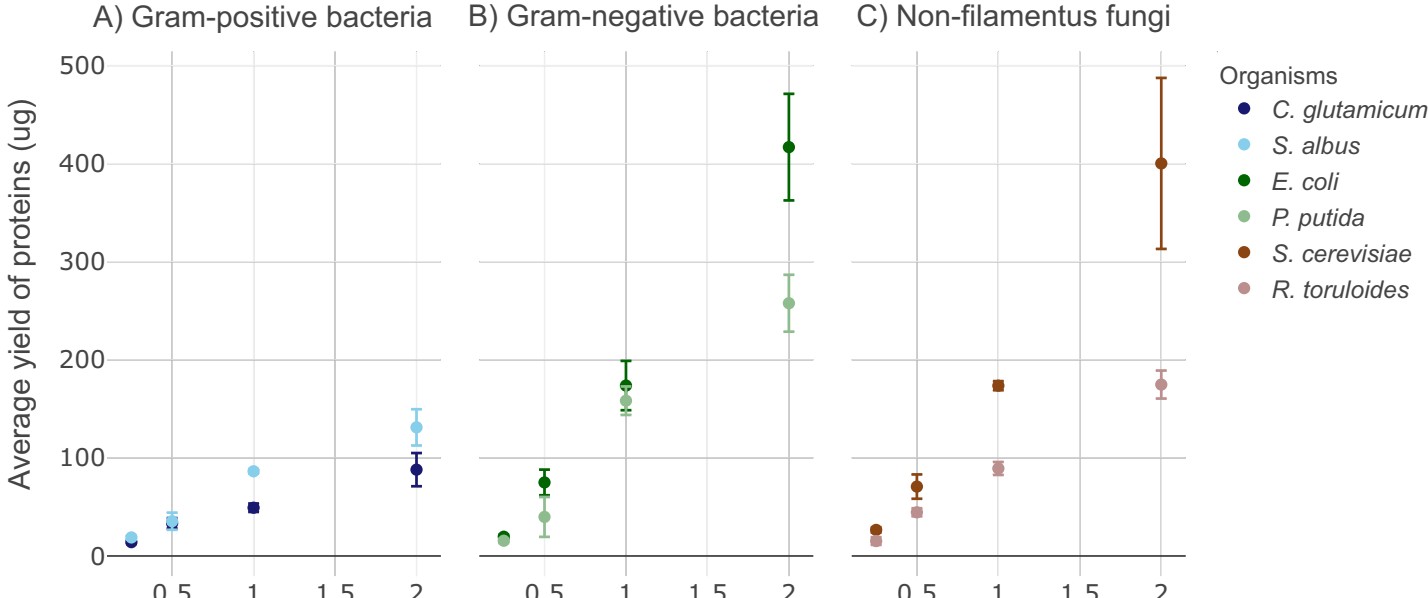

**Fig 1. Scatter plots with data points showing the total protein extracted by using the alkaline-acetone sample preparation protocol on different amounts of biomass from: (A) Gram-positive; (B) Gram-negative; and (C) fungal cells.** The error bars represent three replicates performed on separate days.

OD*mLs cells for gram negative bacteria and non-filamentous fungi species, and 0.5 OD*mLs cells for Gram-positive bacteria with consistent protein recovery (*n* = 4). Sample types other than microbial cell pellets, such as tissues and complex biofluids, haven't been tested with this protocol and may need additional sample preparation steps. Microbes that are highly resistant to cell lysis may require stronger lysis buffer conditions or increased temperatures. Proteolytic digestion of proteins resulting from these samples, however, are readily suitable for LC-MS analysis without the need of additional desalting steps.

Because of its broad applicability, we tested the protocol on mock mixtures of microbial community samples to assess community biomass composition structure using the metaproteomics analysis method established previously [23]. This method estimates the microbial community structure by calculating the percentage of proteinaceous contribution from each member, which is based on the quantitative measurements of the unique peptides that are identified in each member. High-quality, high-throughput, reproducible metaproteomic data will contribute to wider application of such experiments for microbial community analysis [2, 24, 25]. The expected microbial community composition results from samples prepared by the protocol is demonstrated in Figs 2 and 3 by using an Agilent 1290 UHPLC coupled to a Thermo Orbitrap Exploris 480 system operating in data-independent acquisition (DIA) mode [21]. From a LC-MS/MS method (15 minute total run time) we identified more than 1100 proteins and 7000 peptides of each microbe from a 12 μg load of the mock microbial community mixture, consistent with previous applications of the digestion protocols [13]. We prepared mixtures of mock microbial communities consisting of two Gram-negative bacteria (*E. coli* and *P. putida*) or two non-filamentous fungi (*S. cerevisiae* and *R. toruloides*) at four biomass

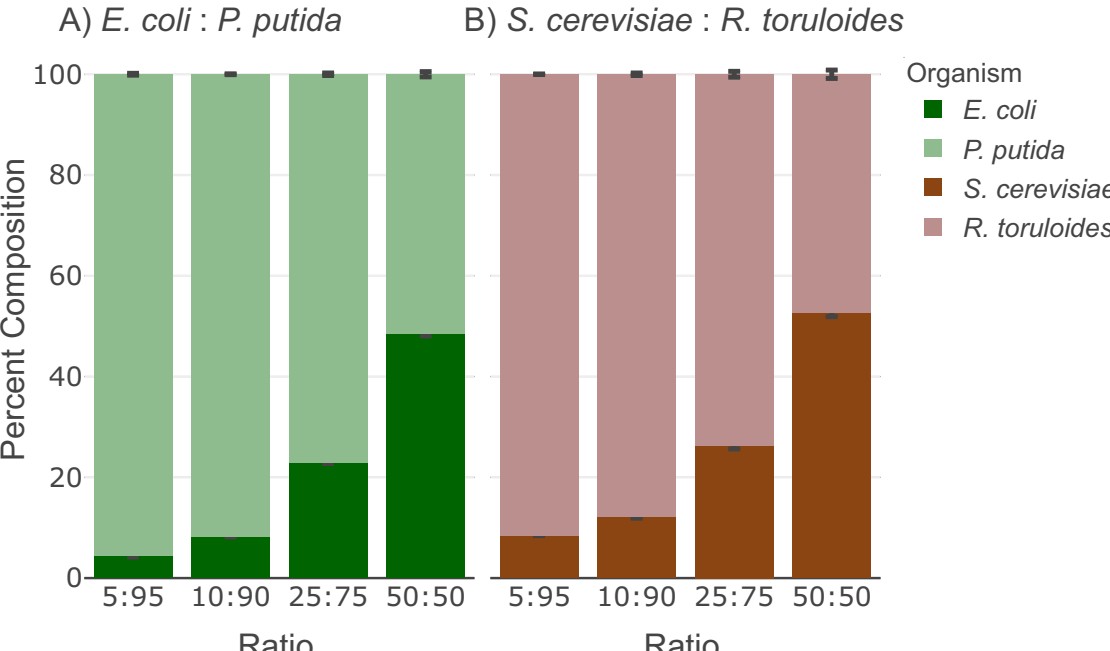

**Fig 2. Community proteomic composition analysis of mock mixtures shows accurate estimation of biomass contribution.** The stacked bar plots display the percentage of biomass contribution from members in (A) Gram-negative mock community, and (B) Non-filamentous fungi mock community. The error bar shows the standard deviation of calculated composition of each member (*n* = 4). The LCMS analysis raw data have been deposited to the ProteomeXchange Consortium data depository at http://www.proteomexchange.org/. They are publicly accessible with the dataset identifier PXD039268.

**Table 1.**

| Mix | % measured | % expected | Mix | % measured | % expected |
|---|---|---|---|---|---|
| *E. coli* | 4.4 ± 0.2 | 5 | *S. cerevisiae* | 8.4 ± 0.1 | 5 |
| *P. putida* | 95.6 ± 0.2 | 95 | *R. toruloides* | 91.6 ± 0.1 | 95 |
| *E. coli* | 8.2 ± 0.1 | 10 | *S. cerevisiae* | 12.1 ± 0.3 | 10 |
| *P. putida* | 92.8 ± 0.1 | 90 | *R. toruloides* | 87.9 ± 0.3 | 90 |
| *E. coli* | 23.0 ± 0.3 | 25 | *S. cerevisiae* | 26.2 ± 0.6 | 25 |
| *P. putida* | 77.0 ± 0.3 | 75 | *R. toruloides* | 73.8 ± 0.6 | 75 |
| *E. coli* | 48.6 ± 0.5 | 50 | *S. cerevisiae* | 52.7 ± 0.8 | 50 |
| *P. putida* | 51.4 ± 0.5 | 50 | *R. toruloides* | 47.3 ± 0.8 | 50 |

ratios. The estimated biomass compositions are reproducible (0.83% standard deviation of the replicates; $n = 4$) and the measured ratios are in close agreement (<3.5%) of the expected ratios as designed (Fig 2, Table 1).

Next, we tested the protocol on a synthetic microbial community of five environmental isolates [22] consisting of one Gram-positive bacterium (ENIGMA isolate: FW305-3-2-15-F-LB2 (GenBank Accession number: OM867407) and four Gram-negative bacteria (ENIGMA isolates: GW456-12-10-14-LB3 (GenBank Accession number: MH795608), GW458-12-9-14-LB2 (GenBank Accession number: MH795601), GW460-LB6 (GenBank Accession number: MH795591), GW460-11-11-14-TSB4 (GenBank Accession number: MH795599)) grown under different experimental conditions to determine composition changes. We started the culture of the five-member microbial community at 20% composition of each and grew them in two culture media, Reasoner's 2A (R2A) [26] and glucose minimal medium.

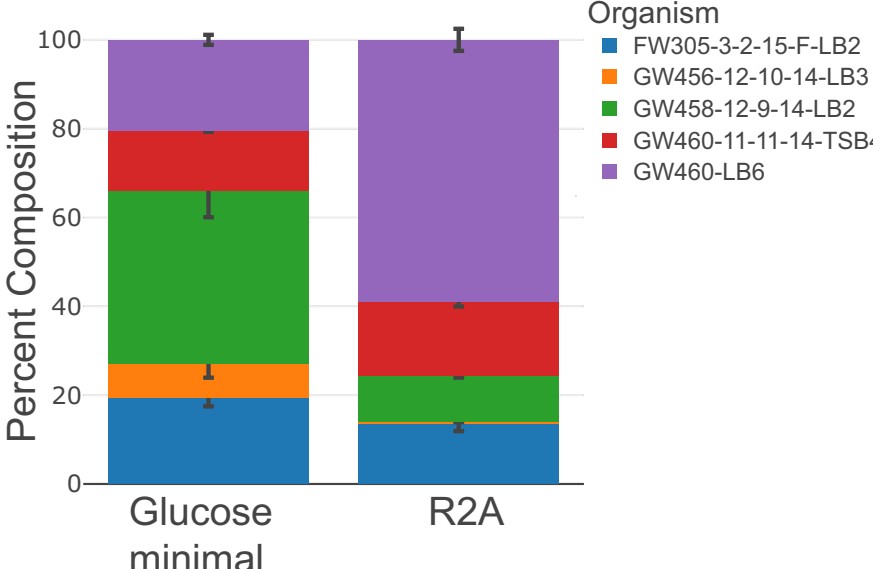

**Fig 3. Community proteomic composition analysis shows accurate estimation of biomass contribution of community structure changes upon different culture conditions.** The stacked bar plots display the percentage of biomass contribution from members of the synthetic microbial community, and the error bars show the standard deviation of calculated composition of each member ($n = 3$). Synthetic microbial community shows distinct structure when culturing in different media, Reasoner's 2A (R2A) and glucose minimal medium. The LCMS analysis raw data have been deposited to the ProteomeXchange Consortium data depository at http://www.proteomexchange.org/. They are publicly accessible with the dataset identifier PXD039268.

We observed significant divergence from the initial composition in each culture medium after 24 hours (Fig 3). The GW456-12-10-14-LB3 isolate decreased in both media while GW458-12-9-14-LB2 increased in glucose minimal medium and GW460-LB6 increased significantly in R2A medium. These differences in community structure reflect the impacts of medium compositions on species interactions and dynamics. This protocol is designed for lab-based, culture conditions and synthetic community experiments where complex sample matrices (*e.g.*, feces, soil) are minimized. Additional sample preservation and/or protein extraction methods may be required for environmental samples to minimize protein degradation and maintain sample integrity. Overall, the flexible design of this protocol enables one researcher to prepare thousands of bottom-up proteomic samples per week. Supporting publications for other monoculture and synthetic community experiments are under development.

## Supporting information

**S1 File. Alkaline-SDS cell lysis with acetone protein precipitation for proteomic sample preparation of microbes in 96-well plate format (PDF).** Also available on protocols.io. (PDF)

## Acknowledgments

The authors thank Dr. Lauren Lui and Dr. Torben Nielsen for help with the proteome information for the ENIGMA organisms. We also thank Dr. Romy Chakraborty for isolating the ENIGMA organisms and Dr. John-Marc Chandonia for help with the genome information sharing of the ENIGMA organisms.

## Author Contributions

**Conceptualization:** Yan Chen, Christopher J. Petzold.

**Data curation:** Yan Chen, Jennifer W. Gin.

**Formal analysis:** Yan Chen, Christopher J. Petzold.

**Funding acquisition:** Nathan J. Hillson, Trent R. Northen, Paul D. Adams.

**Investigation:** Yan Chen, Jennifer W. Gin, Ying Wang, Markus de Raad, Stephen Tan.

**Methodology:** Yan Chen, Jennifer W. Gin.

**Project administration:** Nathan J. Hillson, Trent R. Northen, Paul D. Adams.

**Resources:** Nathan J. Hillson, Trent R. Northen, Paul D. Adams.

**Supervision:** Nathan J. Hillson, Trent R. Northen, Paul D. Adams, Christopher J. Petzold.

**Visualization:** Christopher J. Petzold.

**Writing – original draft:** Yan Chen, Christopher J. Petzold.

**Writing – review & editing:** Yan Chen, Jennifer W. Gin, Ying Wang, Markus de Raad, Stephen Tan, Nathan J. Hillson, Trent R. Northen, Paul D. Adams, Christopher J. Petzold.

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
