## [Decision Letter · Decision Letter 0]

2 May 2023

PONE-D-23-03070

Alkaline-SDS cell lysis of microbes with acetone protein precipitation for proteomic sample preparation in 96-well plate format

PLOS ONE

Dear Dr. Petzold,

Thank you for submitting your manuscript to PLOS ONE. After careful consideration, we feel that it has merit but does not fully meet PLOS ONE’s publication criteria as it currently stands. Therefore, we invite you to submit a revised version of the manuscript that addresses the points raised during the review process.

We look forward to receiving your revised manuscript.

Kind regards,

Faiz ul-Hassan Nasim, PhD

Academic Editor

PLOS ONE

Journal Requirements:

"N.J.H. has financial interests in TeselaGen Biotechnologies, and Ansa Biotechnologies. The other authors have declared that no competing interests exist."

Reviewers' comments:

Reviewer's Responses to Questions

**Comments to the Author**

1. Does the manuscript report a protocol which is of utility to the research community and adds value to the published literature?

Reviewer #1: Yes

Reviewer #2: Yes

2. Has the protocol been described in sufficient detail?

To answer this question, please click the link to protocols.io in the Materials and Methods section of the manuscript (if a link has been provided) or consult the step-by-step protocol in the Supporting Information files.

The step-by-step protocol should contain sufficient detail for another researcher to be able to reproduce all experiments and analyses.

Reviewer #1: Yes

Reviewer #2: No

3. Does the protocol describe a validated method?

Reviewer #1: Yes

Reviewer #2: Yes

4. If the manuscript contains new data, have the authors made this data fully available?

Reviewer #1: Yes

Reviewer #2: Yes

**5. Is the article presented in an intelligible fashion and written in standard English?**

Reviewer #1: Yes

Reviewer #2: Yes

6. Review Comments to the Author

Reviewer #1: Comments:

Abstract: Authors should be refurnished abstract.

1.Authors should be checked grammar “in alkaline chemical buffer” or “in an alkaline chemical buffer”?

Introduction: Authors should be rechecked

Result sections:

1.You have extracted protein from Gram-positive bacteria species, Gram-negative bacteria species, and non-filamentous fungi species with ~2x109 cells of biomass from each species but why did Gram-positive bacteria species total protein obtained very less than Gram-negative species and non-filamentous?. Does protein extraction concentration depend on species? Any specification on the extraction?

2.Haven’t tested any animal cell line/tissues. You should test on primary cells or cell lines because that is good for justification for microbial protocol. The author should at least do the test on either animal tissues/primary cells/cell lines. It may be not applicable to animal cells.

Figure Captions:

1.Fig.1: Error bars should be represented by a minimum of three replicates performed on a separate day. Any justification for why the authors' error bars represented two replicates performed/experiments?

Reviewer #2: Although the authors have given the reference of the protocol in materials and method section, It will be better to describe the protocol here as well. So that It will make the draft convenient for the readers to assess the total information at one place.

The results mentioned were actual results so heading must be changed from expected to more suitable word, or Just 'Results'.

Please elaborate the statement ' With two OD*mL (~2 ×109 cells) of biomass,' in the expected results section, as it is not clear.

7. PLOS authors have the option to publish the peer review history of their article (what does this mean?). If published, this will include your full peer review and any attached files.

Reviewer #1: **Yes: **Dr. Aabid Hussain

Reviewer #2: **Yes: **Dr. Adnan Akhter

---

## [Author Response · Author response to Decision Letter 0]

8 Jun 2023

Response to Reviewers:

Manuscript Number PONE-D-23-03070

Alkaline-SDS cell lysis of microbes with acetone protein precipitation for proteomic sample preparation in 96-well plate format

Recommendation: Publish after resubmission. Comments:

Abstract: Authors should be refurnished abstract.

1. Authors should be checked grammar “in alkaline chemical buffer” or “in an alkaline chemical buffer”?

--We corrected the grammar as the reviewer suggested.

Introduction: Authors should be rechecked Result sections:

1. You have extracted protein from Gram-positive bacteria species, Gram-negative bacteria species, and non-filamentous fungi species with ~2x109 cells of biomass from each species but why did Gram-positive bacteria species total protein obtained very less than Gram-negative species and non-filamentous?. Does protein extraction concentration depend on species? Any specification on the extraction?

--We thank the reviewer for the question. Yes, the total protein obtained varies depending on the species. Gram-positive bacteria have a thicker cell wall than Gram-negative bacteria, consequently, the percentage of protein is lower for the Gram-positive bacteria on a per cell basis and, correspondingly, for the OD measurements. We clarified this point in the text by adding this statement:

“The amount of protein extracted from the microbes will depend on the species of interest, consequently the starting amount of biomass may need to be adjusted for a given application.” 

2. Haven’t tested any animal cell line/tissues. You should test on primary cells or cell lines because that is good for justification for microbial protocol. The author should at least do the test on either animal tissues/primary cells/cell lines. It may be not applicable to animal cells.

--We thank the reviewer for the suggestion, however, testing on animal cells is beyond the scope of our research and our available resources. This protocol has been tested and applied for microbe analysis because that is the focus of our research, but the reviewer is correct that it may not be applicable to animal cells. 

Figure Captions:

1. Fig.1: Error bars should be represented by a minimum of three replicates performed on a separate day. Any justification for why the authors' error bars represented two replicates performed/experiments?

--For the protein quantification measurements, we decided to examine a larger number of conditions at the expense of the number of replicates. We recollected the data with three replicates to alleviate this concern. Figure 1 is updated with the triplicate measurements and the values have been updated in the text.

---

## [Editor Report · Decision Letter 1]

20 Jun 2023

Alkaline-SDS cell lysis of microbes with acetone protein precipitation for proteomic sample preparation in 96-well plate format

PONE-D-23-03070R1

Dear Dr. Petzold,

We’re pleased to inform you that your manuscript has been judged scientifically suitable for publication and will be formally accepted for publication once it meets all outstanding technical requirements.

Kind regards,

Faiz ul-Hassan Nasim, PhD

Academic Editor

PLOS ONE
---

## [Editor Report · Acceptance letter]

28 Jun 2023

PONE-D-23-03070R1 

Alkaline-SDS cell lysis of microbes with acetone protein precipitation for proteomic sample preparation in 96-well plate format 

Dear Dr. Petzold:

I'm pleased to inform you that your manuscript has been deemed suitable for publication in PLOS ONE. Congratulations! Your manuscript is now with our production department. 

Kind regards, 

on behalf of

Dr. Faiz ul-Hassan Nasim 

Academic Editor

PLOS ONE